# Evaluation of the horizontal approach to the medial malleolar facet in sagittal talar fractures through dorsiflexion and plantarflexion positions

Xian Li[1,2,3☯], Xiao-ke Wang[3☯], Li-ren Han[1], Hao Li[1], Hui-chao Tian[1], Jun Yan[1,2]*, Hai-juan Liu[4]*

1 Department of Orthopaedic Surgery, Liaocheng People's Hospital, Liaocheng, China, 2 Beijing Jishuitan Hospital Liaocheng Hospital, Liaocheng, China, 3 School of Clinical Medicine, Shandong Second Medical University, Weifang, China, 4 Department of Endocrinology, Liaocheng People's Hospital, Clinical Hospital of Shandong First Medical University, Liaocheng, China

☯ These authors contributed equally to this work.
* awpkaka@126.com (JY); emilyeffi@163.com (HJL)

**Data Availability Statement:** All relevant datasets for this study are publicly available from the Dryad

## Abstract

### Background

Talar fractures often require osteotomy during surgery to achieve reduction and screw fixation of the fractured fragments due to limited visualization and operating space of the talar articular surface. The objective of this study was to evaluate the horizontal approach to the medial malleolus facet by maximizing exposure through dorsiflexion and plantarflexion positions.

### Methods

In dorsiflexion, plantarflexion, and functional foot positions, we respectively obtained the anterior and posterior edge lines of the projection of the medial malleolus on the medial malleolar facet. The talar model from Mimics was imported into Geomagic software for image refinement. Then Solidworks software was used to segment the medial surface of the talus and extend the edge lines from the three positions to project them onto the "semicircular" base for 2D projection. The exposed area in different positions, the percentage of total area it represents, and the anatomic location of the insertion point at the groove between the anteroposternal protrusions of the medial malleolus were calculated.

### Results

The mean total area of the "semicircular" region on the medial malleolus surface of the talus was 542.10 ± 80.05 mm$^2$. In the functional position, the exposed mean area of the medial malleolar facet around the medial malleolus both anteriorly and posteriorly was 141.22 ± 24.34 mm$^2$, 167.58 ± 22.36mm$^2$, respectively. In dorsiflexion, the mean area of the posterior aspect of the medial malleolar facet was 366.28 ± 48.12 mm$^2$. In plantarflexion, the mean of the anterior aspect of the medial malleolar facet was 222.70 ± 35.32 mm$^2$. The mean

repository (https://doi.org/10.5061/dryad.r7sqv9skk).

**Funding:** This study was financially supported by the National Natural Science Foundation of ShanDong Province in the form of a grant (ZR202102280280) received by JY. The grant funds allocated for this study were specifically used to collect CT scans. No additional external funding was received for this study.

**Competing interests:** The authors have declared that no competing interests exist.

overlap area of unexposed area in both dorsiflexion and plantarflexion was $23.32 \pm 5.94$ mm$^2$. The mean percentage of the increased exposure area in dorsiflexion and plantarflexion were $36.71 \pm 3.25\%$ and $15.13 \pm 2.83\%$. The mean distance from the insertion point to the top of the talar dome was $10.69 \pm 1.24$ mm, to the medial malleolus facet border of the talar trochlea was $5.61 \pm 0.96$ mm, and to the tuberosity of the posterior tibiotalar portion of the deltoid ligament complex was $4.53 \pm 0.64$ mm.

## Conclusions

Within the 3D model, we measured the exposed area of the medial malleolus facet in different positions and the anatomic location of the insertion point at the medial malleolus groove. When the foot is in plantarflexion or dorsiflexion, a sufficiently large area and operating space can be exposed during surgery. The data regarding the exposed visualization area and virtual screws need to be combined with clinical experience for safer reduction and fixation of fracture fragments. Further validation of its intraoperative feasibility will require additional clinical research.

## Introduction

Talar fractures are relatively uncommon, accounting for less than 1% of all fractures, and comprising only 3% to 6% of fractures in the foot and ankle [1]. The talus is composed of a head, neck, and body, with an extremely irregular shape, and the majority of it is covered by articular cartilage [2]. Most talar fractures are caused by violent trauma such as falls from height or car accidents [3]. Furthermore, talar fractures are often associated with ankle fractures, severe soft tissue damage, calcaneal fractures, and metatarsal fractures. These can disrupt the ankle joint's mobility system and lead to complications such as ischemic necrosis, arthritis and deformities [3–7]. Therefore, talar fractures should be stabilized, anatomically reduced, and firmly fixed as early as possible [8, 9].

Talar body fractures account for 7% to 38% of all talar fractures and are one of the most challenging foot and ankle procedures [7, 10]. The structure of the talar body is unique, its upper surface connects to the tibial plafond, and it extends laterally to attach to the medial and lateral malleoli [2, 5]. Such complex anatomy hinders the exposure, reduction, and fixation of the fractured fragments during surgery [11]. Previous studies have typically employed osteotomy of the medial or lateral malleolus in the treatment of talar body fractures to maximize the exposure of the talus' upper articular surface, especially in cases of comminuted and displaced fractures [4, 7, 12, 13]. However, in our surgical experience with talar procedures, we have found that when the foot is maximally dorsiflexed or plantarflexed, it is possible to expose a significant portion of the medial malleolar facet. In some cases of talar body sagittal fractures, screw fixation may be achieved without the need for osteotomy. This study was designed to evaluate the horizontal approach to the medial malleolus surface in sagittal talar fractures through dorsiflexion and plantarflexion.

## Materials and methods

We prospectively recruited 91 adults patients who underwent postoperative follow-up for unilateral lower limb fractures. Dual lower limb consecutive CT scans were performed, with the unaffected side foot maintaining functional position, maximal dorsiflexion, and maximal plantarflexion, each undergoing one CT scan at the imaging research center of our hospital during

December 2022 and June 2023, including 38 females and 53 males. Patients were excluded if they had deformity, fracture, arthritis, or tumor in the foot. This study was conducted in accordance with the World Medical Association Declaration of Helsinki and approved by the Ethical Committee of Liaocheng People's Hospital (No. 2021033). Written informed consent was obtained from all patient who participated in this study (includes patient with traction as shown in Fig 4). The mean age of the patients on whom the models were based was 41.47 ± 14.98 years (range, 18–74 years).

DICOM-formatted CT-scan images of each patient were imported into Mimics software (21.0; Materialise, Leuven, Belgium). We removed the soft tissue and affected bones by the function of image segmentation, region growth and multiple slice editing of Mimics software, respectively. A total of 273 virtual foot and ankle models were created. In dorsiflexion, plantarflexion, and functional foot positions, we respectively obtained the anterior and posterior edge lines of the projection of the medial malleolus on the medial malleolar facet (Fig 1). At the same time, we found that the medial malleolar facet had a shape resembling a "semicircle", and regardless of whether the foot was in the functional position, dorsiflexed, or plantarflexed, the movement of the medial malleolus occurred within this "semicircular" region (Fig 2A). Tracing the outline of the foot in the three positions on the talus, it is not difficult to observe that plantarflexion and dorsiflexion expand the exposure area on the anterior and posterior aspects of the talus (Fig 2B). The talar model from Mimics was imported into Geomagic software (2019, 3D Systems, North Carolina, USA) for image refinement. Then Solidworks software (2021, Dassault Systemes, USA) was used to segment the medial surface of the talus and

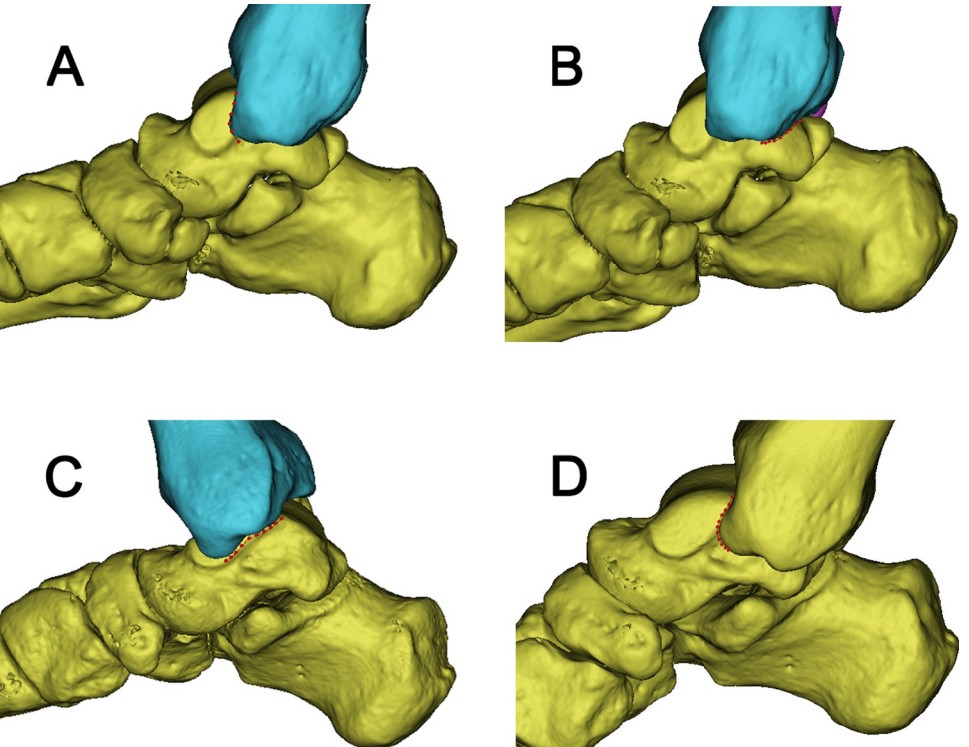

**Fig 1. The anterior and posterior edge lines of the projection of the medial malleolus on the medial surface of the talus (red lines). A** In the functional position, the anterior edge lines of the projection of the medial malleolus. **B** In the functional position, the posterior edge lines of the projection of the medial malleolus. **C** In dorsiflexion, the posterior edge lines of the projection of the medial malleolus. **D** In plantarflexion, the anterior edge lines of the projection of the medial malleolus.

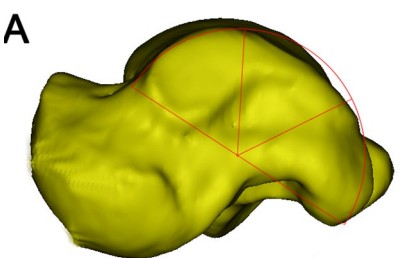
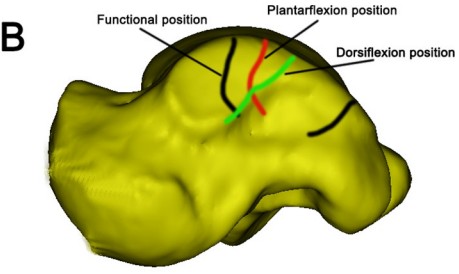

**Fig 2. The measurement of exposed area of medial malleolar facet. A** The "semicircular" area of the medial malleolar facet. **B** The projected lines on the medial malleolus facet in the three positions (Black-Functional position; Red-Plantarflexion position; Green-Dorsiflexion position).

extend the edge lines from the three positions to project them onto the "semicircular" base for 2D projection. The following areas were measured: Total area (TA) of the "semicircular" region on the medial malleolar facet; In the functional position, the exposed area of the medial malleolar facet around the medial malleolus both anteriorly (FAA) and posteriorly (FAP); In dorsiflexion, the exposed area of the posterior aspect of the medial malleolar facet (DA); In plantarflexion, the exposed area of the anterior aspect of the medial malleolar facet (PA); Unexposed overlap area in both dorsiflexion and plantarflexion (UA). The percentage of the increased exposure area in dorsiflexion and plantarflexion positions were calculated.

In addition, in the functional position, we observed that the groove between the anteroposternal protrusions of the medial malleolus was closer to the center of the medial malleolar facet. This groove could potentially serve as the insertion point for screw fixation (Fig 3A). To determine the anatomic location of the insertion point on the medial malleolar facet, a virtual screw was inserted into the talar model and the distances were measured: the distance of the insertion point to the top of the talar dome (L1), the distance of the medial malleolus facet border of the talar trochlea (red line, L2), and the distance from the insertion point to the tuberosity of the posterior tibiotalar portion of the deltoid ligament complex (L3), respectively (Fig 3B).

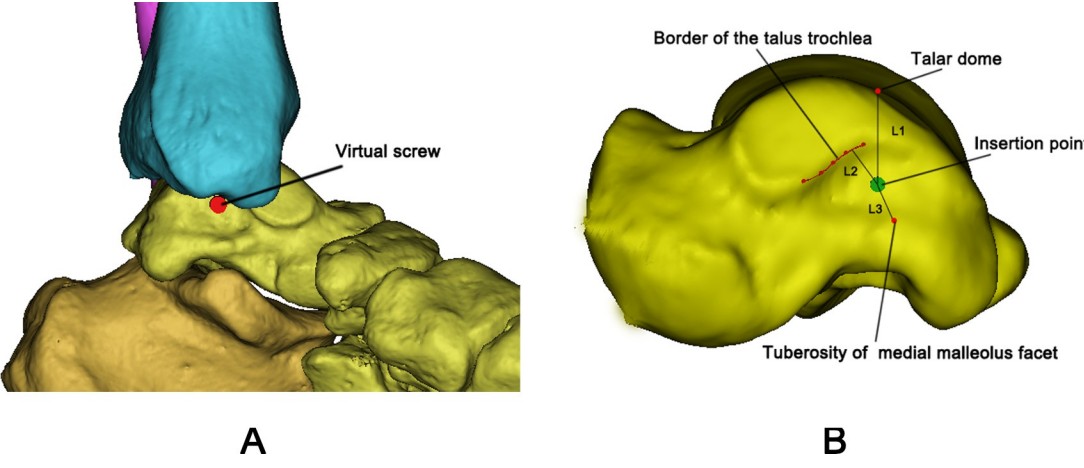

**Fig 3. The measurement of virtual screws in the model. A** The insertion point of the virtual screw is within the groove between the anteroposternal protrusions of the medial malleolus. **B** Anatomic location of virtual screw (L1: the distance of the insertion point to the top of the talar dome; L2: the distance of the insertion point to the medial malleolus facet border of the talus trochlea; L3: the distance of the insertion point to the tuberosity of the posterior tibiotalar portion of the deltoid ligament complex).

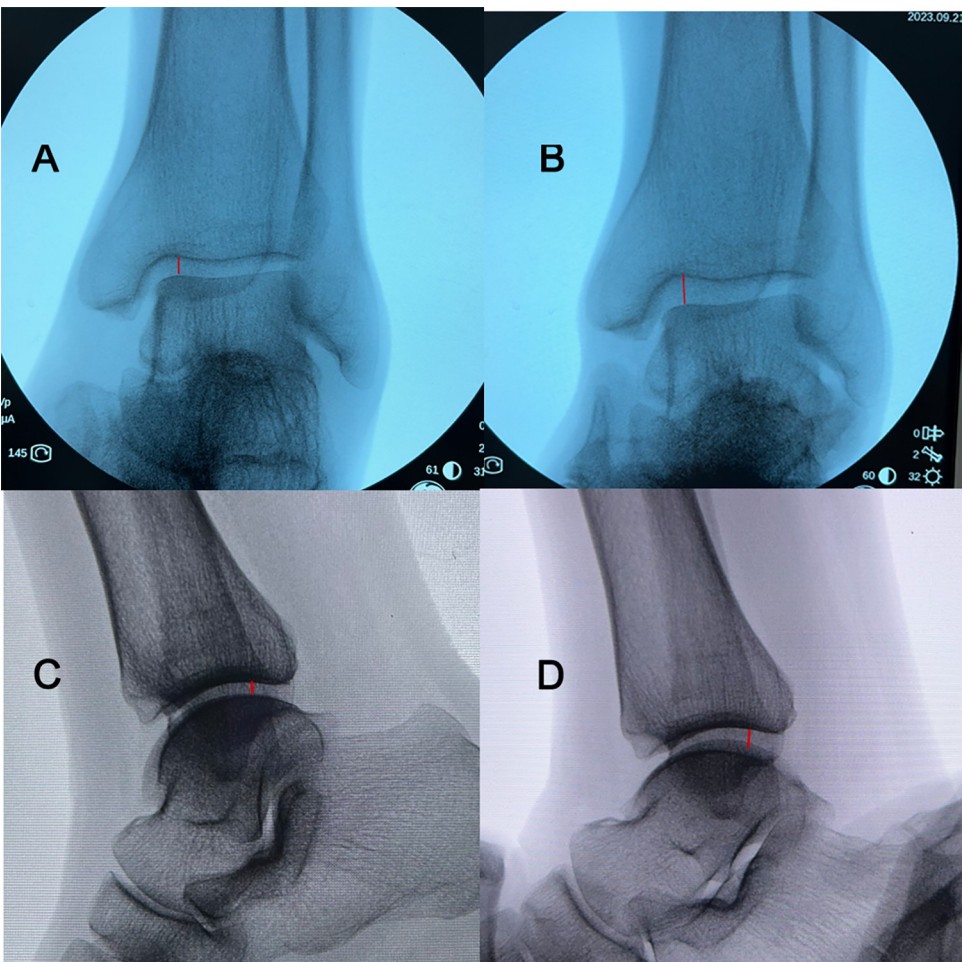

**Fig 4. After applying traction, the joint space widens (red lines). A** In the neutral position (no traction). **B** In the neutral position (with traction). **C** In the lateral position (no traction). **D** In the lateral position (with traction).

The collected data were analysed by SPSS 25.0 statistical software. The experimental data are represented as the mean ± standard deviation. The independent sample T test was used to compare the genders. Statistical significance was accepted at $P < 0.05$.

## Results

The study subjects included 53 males and 38 females aged between 18 and 74 years old, with a mean age of 41.47 ± 14.98 years.

As shown in Table 1, the mean total area of the "semicircular" region on the medial surface of the talus was 542.10 ± 80.05 mm$^2$; In the functional position, the exposed mean area of the medial surface of the talus around the medial malleolus both anteriorly and posteriorly was 141.22 ± 24.34 mm$^2$, 167.58 ± 22.36mm$^2$. For the data captured above, the exposed mean area (TA, FAA, FAP) of the intersex difference was significant (P<0.05). In dorsiflexion, the mean area of DA was 366.28 ± 48.12 mm$^2$; In plantarflexion, the mean of PA was 222.70 ± 35.32 mm$^2$; The mean area of UA was 23.32 ± 5.94 mm$^2$. For the data captured above, the exposed mean area (DA, PA, UA) of the intersex difference was significant (P<0.05). However, in dorsiflexion, the mean percentage of the increased exposure area was 36.71 ± 3.25%, and in

**Table 1. Comparison between different genders: Average TA, FAA, FAP, DA, PA, UA, (DA-FAP)/TA and (PA-FAA)/TA.**

| Group | TA[y] (mm²) | FAA[y] (mm²) | FAP[y] (mm²) | DA[y] (mm²) | (DA-FAP)/TA (%) | PA[y] (mm²) | (PA-FAA)/TA (%) | UA[y] (mm²) |
|---|---|---|---|---|---|---|---|---|
| All (n = 91) | 542.10±80.05 | 141.22±24.34 | 167.58±22.36 | 366.28±48.12 | 36.71±3.25 | 222.70±35.32 | 15.13±2.83 | 23.32±5.94 |
| Male (n = 53) | 599.37±68.81 | 158.62±20.39 | 179.29±23.63 | 399.13±49.11 | 36.51±3.26 | 248.12±33.88 | 14.94±3.25 | 28.511±4.23 |
| Female (n = 38) | 490.03±47.89 | 125.39±15.25 | 156.95±15.38 | 336.42±20.03 | 36.90±3.40 | 199.60±15.12 | 15.31±2.53 | 18.60±1.73 |
| t value[x] | 4.261 | 4.256 | 2.593 | 3.902 | -0.262 | 4.310 | -0.292 | 7.155 |
| P value[x] | 0.001 | <0.001 | 0.018 | 0.001 | 0.796 | <0.001 | 0.773 | <0.001 |

[x]t and P are the results of gender comparisons

[y]For the area of TA, FAA, FAP, DA, PA, and UA intersex difference was significant ($P < 0.05$)

plantarflexion, the mean percentage of the increased exposure area was 15.13 ± 2.83%, fom Table 1, the results were not statistically significant between males and females (P>0.05).

From Table 2, the mean distance L1 was 10.69 ± 1.24 mm, L2 was 5.61 ± 0.96 mm, and L3 was 4.53 ± 0.64 mm, respectively. For the data captured above, the distance (L1, L2, L3) of the intersex difference was significant (P<0.05).

## Discussion

In our research, the areas of the TA, FAA, FAP, DA, PA, UA were significantly larger in males compared with females. This may be due to anatomical differences in the bone size between females and males [14, 15]. From Table 1, The exposed area on the medial malleolar surface of the talus significantly increases when the foot was in dorsiflexion or plantarflexion, and the increased area in plantarflexion accounted for one-third of the total talar area. In conjunction with Fig 1, when there is a sagittal fracture of the talar body, it is entirely feasible to use screws to fix the fracture fragments separately in dorsiflexion and plantarflexion. The extensive exposure in dorsiflexion provides a sufficient distance from the deltoid ligament complex, the artery of the tarsal canal, the posterior tibial artery, and tibial nerve when performing screw fixation, ensuring an ample and safe operating space [4]. As shown in Fig 1, In dorsiflexion and plantarflexion, the joint becomes locked, increasing the contact area of the joint and providing ample exposure of the talus articular surface. Increased exposure, sufficient space, and joint locking; these factors are advantageous for the reduction of displaced fracture fragments. For fracture patients, there may be damage to the deltoid ligament complex the tibialis anterior, the tibialis posterior, the flexor hallucis longus, and the flexor digitorum longus [3, 10]. Additionally, muscle relaxation under anesthesia can lead to a larger surgical exposure area, which is beneficial for improving the visualization of the fracture fragments [16]. DeKeyser et al demonstrated that the posterior medial and posterior lateral approaches can increase the

**Table 2. Comparison between different genders: Average $L_1$, $L_2$, $L_3$.**

| Group | $L_1$[y] (mm) | $L_2$[y] (mm) | $L_3$[y] (mm) |
|---|---|---|---|
| All (n = 91) | 10.69±1.24 | 5.61±0.96 | 4.53±0.64 |
| Male (n = 53) | 11.39±1.20 | 6.16±0.81 | 4.88±0.62 |
| Female (n = 38) | 10.06±0.91 | 5.11±0.82 | 4.21±0.49 |
| t value[x] | 2.863 | 2.926 | 2.724 |
| P value[x] | 0.010 | 0.009 | 0.013 |

[x]t and P are the results of gender comparisons

[y]For the distance of $L_1$, $L_2$ and $L_3$, intersex difference was significant ($P < 0.05$)

exposure area of the talar dome with traction [17, 18]. Therefore, applying traction during the operation can further increase the operating space. As shown in Fig 2B and Table 1, the area not exposed in dorsiflexion and plantarflexion are relatively small, and will further diminish or even disappear during traction.

From Fig 3B, we found that the insertion point at the groove between the anteroposternal protrusions of the medial malleolus is not directly beneath the highest point of the talar dome. However, it is within the space exposed during dorsiflexion. When visualizing screw fixation at this location is limited in the functional position, increasing the operating space can be achieved by transitioning to dorsiflexion. During the operation, under lead shielding for protection, two operators manually applied traction to the talus in both the neutral position and lateral position. It can be observed that the joint space between the talus and tibia widens, and the insertion point also will moves closer to the center of the talar body accordingly (Fig 4). Building upon the research of DeKeyser et al, applying posterior-inferior traction during dorsiflexion will further expand the visualization area, allowing for screw placement closer to the center of the talar body to achieve maximum holding force for fixing the fractured fragments. As shown from Table 2, the distances of the L1, L2, L3 were significantly larger in males compared with females. This may be due to anatomical differences in the bone size between females and males [14, 15].

This study evaluated the horizontal approach to the medial malleolus surface in sagittal talar fractures through dorsiflexion and plantarflexion. The findings demonstrate a significant increase in the exposure area of the medial malleolus facet in both dorsiflexion and plantarflexion positions. Moreover, under anesthesia and traction during surgery, there is a notable enhancement in the visualization of the talus. This may facilitate the reduction and screw fixation of fracture fragments without osteotomy, thereby reducing patient trauma associated with the surgery.

There are still some limitations to this study. We only analyzed the data based on genders, not different age, body size and type of fracture groups. These factors may also affect the visualization of the medial malleolus facet and the positioning of screws. In addition, the study was performed on intact tali, but the normal anatomy is distorted in the case of talar fractures. It is necessary to improve the quality of fracture reduction by preoperative 3D reconstruction and intraoperative use of reduction forceps. Furthermore, we lack a sufficient number of cadaveric specimens to verify the exposed area of the medial malleolus facet in different positions and the operational space for reduction and screw placement during surgery. More relevant clinical studies are needed to confirm these findings.

## Conclusion

We provide guidelines for the horizontal approach to the medial malleolus facet in sagittal talar fractures in a 3D simulation. Within the 3D model, we measured the exposed area of the medial malleolus facet in different positions and the anatomic location of the insertion point at the medial malleolus groove. The data regarding the exposed visualization area and virtual screws need to be combined with clinical experience for safer reduction and fixation of fracture fragments. Further validation of its intraoperative feasibility will require additional cadaveric specimens research and clinical studies.

## Supporting information

**S1 Data. Minimum data set.**
(DOCX)

## Author Contributions

**Conceptualization:** Xian Li.

**Data curation:** Xiao-ke Wang.

**Investigation:** Hao Li.

**Project administration:** Jun Yan.

**Resources:** Li-ren Han.

**Software:** Hui-chao Tian.

**Supervision:** Hai-juan Liu.

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
