## [Decision Letter · Decision Letter 0]

14 Dec 2023

PONE-D-23-36659Evaluation of the horizontal approach to the medial malleolar facet in sagittal talar fractures through dorsiflexion and plantarflexion positions.PLOS ONE

Dear Dr. Yan,

Thank you for submitting your manuscript to PLOS ONE. After careful consideration, we feel that it has merit but does not fully meet PLOS ONE’s publication criteria as it currently stands. Therefore, we invite you to submit a revised version of the manuscript that addresses the points raised during the review process.

We look forward to receiving your revised manuscript.

Kind regards,

Amir Human Hoveidaei

Academic Editor

PLOS ONE

Journal Requirements:

3. Thank you for stating the following in the Acknowledgments Section of your manuscript: "This study was supported by the National Natural Science Foundation of ShanDong province (No. ZR202102280280)."

Please remove any funding-related text from the manuscript and let us know how you would like to update your Funding Statement. Currently, your Funding Statement reads as follows: "The authors received no specific funding for this work."

Additional Editor Comments:

Dear authors,

Many thanks for your submission.

Please perform the required revisions based on the reviewer comments before acceptance.

Reviewers' comments:

Reviewer's Responses to Questions

**Comments to the Author**

1. Is the manuscript technically sound, and do the data support the conclusions?

Reviewer #1: Yes

Reviewer #2: Yes

2. Has the statistical analysis been performed appropriately and rigorously? 

Reviewer #1: Yes

Reviewer #2: Yes

3. Have the authors made all data underlying the findings in their manuscript fully available?

Reviewer #1: Yes

Reviewer #2: Yes

4. Is the manuscript presented in an intelligible fashion and written in standard English?

Reviewer #1: Yes

Reviewer #2: Yes

5. Review Comments to the Author

Reviewer #1: Thank you for the opportunity to review this interesting article on medial malleolar approaches to sagittal talar fractures. This seems to be a novel addition to the literature with impressive methods and results.

There are many grammatical/punctuation errors which will need to be addressed prior to publication. Please revise semicolon usage. Also, “in plantarflexion” is correct whereas “in plantarflexion position” is not. “In the plantarflexion position” is correct as well.

Please find my revisions below with corresponding line numbers:

33-35: change to “The talar model from Mimics was imported into Geomagic software for image refinement. Then Solidworks software was used to segment the medial surface of the talus and extend the 35 edge lines from the three positions to project them onto the "semicircular" base for 2D projection”. This can be edited for clarity as well.

39-50: Replace semicolons with periods (check for whole section for punctuation errors)

42: add “, respectively”

42: remove “position”

62: please standardize either “foot-and-ankle” or “foot and ankle” throughout the manuscript

64: should be “falls”

68: please elaborate on “etc” or remove

68: add comma after reduced

71: delete “surgeries in”

78: The foot cannot be dorsiflexed and plantarflexed at the same time. Please clarify - replace “and” with “or”

87: were patients retrospectively or prospectively recruited? Please clarify

94: Was consent obtained for all patients? Or just the one whose radiographs are in figure 4? If all patients please updated the text. If only figure 4 please change the text to explain it that it is the one patient whose radiographs are displayed in the manuscript.

95: what does “all methods were carried out in accordance with relevant guidelines”? Which methods and what guidelines?

96: should be formatted “(range, 18-74 years)” please correct for the rest of the manuscript

108-111: make changes suggested in my comment for lines 33-35

119: should be “in the functional position” correct if this error is found elsewhere in the manuscript

123: replace “measured the distances” with “the distances were measured”

124: write “the distance of the medial malleolar facet border …” … “(red line, L2), and the distance from the insertion point to the tuberosity of the medial malleolus facet …”

125: put the L1, L2, L3 as (L1) after the description. Ex: “from the insertion point to the top of the talar dome (L1), …”

125: I don’t think I’ve seen this wording before: tuberosity of medial malleolus facet of deltoid ligament posterior tibial talar. Please explain and re-word. Is this the posterior tibiotalar portion of the deltoid ligament complex? (Change this phrasing in wherever else it is applicable ex: Fig 3)

128: write out standard deviation instead of “SD”

128: what kind of T test? Student t-test? Wilcoxin sign-rank? T tests were used to compare what data?

138, 141, 146: in what manner was the intersex difference significant? Please elaborate. Intersex difference in area?

152-153: needs citation. Also, what are the differences between female and male? Also it should be “females and males” in the text

157-159: needs citation. What tendons, nerves, and vessels may be in the way.

161: which factors? Please elaborate

162-164: need citations. which tendons? Which muscles? Provide study citation for muscle relaxation leading to greater exposure area.

174-175: informed consent statement disrupts flow of discussion, remove (assuring that it can be found in the methods section)

181: replace obviously - too informal

182-183: As with 152-153, please describe the differences with literature. Also it should be “females and males” in the text

188-190: please cite literature on the reduced economic burden

193-194: please cite literature to support your claim that “Chinese people … have different skeletal shapes with American and European populations.” Also it should be different shapes “than”

195: plural of talus is tali, please update text

196-197: please elaborate on what is meant by “It is necessary to improve the quality of fracture reduction by preoperative 3D reconstruction and intraoperative use of reduction forceps.” How does this fit into the limitations section?

Reviewer #2: This study presents a commendable and insightful approach to a complex subject. However, I would like to offer some suggestions to enhance the clarity and impact of the paper.

Abstract:

Line 46: This sentence requires correction for better clarity. Additionally, it would be beneficial to include information about UA in the abstract.

Line 51: In the conclusion part of the abstract, it's essential to explicitly mention the findings related to your horizontal approach, as it is central to the study.

Introduction:

The introduction is well-written and sets a strong foundation for the study.

Methods:

Consider organizing the Methods section into clearly defined subsections. This would greatly enhance the readability and logical flow of the paper.

Some content in the Methods section appears to pertain to the Results. Please review and relocate these sentences to the appropriate section.

Results:

For a more comprehensive analysis, consider using the proportion of each variable relative to the total talus surface area. This adjustment could account for variations in talus size due to factors like height and gender.

Lines 141-144 require rewriting for better clarity and coherence.

Discussion:

It would be advantageous to start the discussion with the study's most significant findings, as this sets the tone and focus for the rest of the discussion.

The section discussing limitations is well-articulated.

Conclusion:

The conclusion is well-formed and effectively summarizes the study.

6. PLOS authors have the option to publish the peer review history of their article (what does this mean?). If published, this will include your full peer review and any attached files.

Reviewer #1: No

Reviewer #2: **Yes: **Mohammad Poursalehian

---

## [Author Response · Author response to Decision Letter 0]

13 Jan 2024

"Response to Reviewers"has been submitted

---

## [Decision Letter · Decision Letter 1]

26 Feb 2024

PONE-D-23-36659R1Evaluation of the horizontal approach to the medial malleolar facet in sagittal talar fractures through dorsiflexion and plantarflexion positions.PLOS ONE

Dear Dr. Yan,

Thank you for submitting your manuscript to PLOS ONE. After careful consideration, we feel that it has merit but does not fully meet PLOS ONE’s publication criteria as it currently stands. Therefore, we invite you to submit a revised version of the manuscript that addresses the points raised during the review process.

We look forward to receiving your revised manuscript.

Kind regards,

Amir Human Hoveidaei

Academic Editor

PLOS ONE

Journal Requirements:

Additional Editor Comments:

Dear authors,

Many thanks for submitting the paper. Based on the reviewer comments we can accept after minor revisions.

Best,

Amir H Hoveidaei, MD, MSc

Reviewers' comments:

Reviewer's Responses to Questions

**Comments to the Author**

1. If the authors have adequately addressed your comments raised in a previous round of review and you feel that this manuscript is now acceptable for publication, you may indicate that here to bypass the “Comments to the Author” section, enter your conflict of interest statement in the “Confidential to Editor” section, and submit your "Accept" recommendation.

Reviewer #1: (No Response)

Reviewer #2: All comments have been addressed

2. Is the manuscript technically sound, and do the data support the conclusions?

Reviewer #1: Yes

Reviewer #2: Yes

3. Has the statistical analysis been performed appropriately and rigorously? 

Reviewer #1: Yes

Reviewer #2: Yes

4. Have the authors made all data underlying the findings in their manuscript fully available?

Reviewer #1: Yes

Reviewer #2: Yes

5. Is the manuscript presented in an intelligible fashion and written in standard English?

Reviewer #1: Yes

Reviewer #2: Yes

6. Review Comments to the Author

Reviewer #1: Thank you for the opportunity to once again review this compelling article. At this time I believe that there remains some minor revisions before this article is acceptable for publication. Please find below my review.

Whole article: There should be spaces before parenthesis “ (“

Abstract:

34: change to “In dorsiflexion, plantarflexion, and functional foot positions” and make sure this grammatical correction is applied to other instances. On line 110 “in the functional position, dorsiflexed, or plantarflexed” is also an acceptable way to phrase this.

36: delete extra period

Introduction:

67: change all instances of “foot-and-ankle” to “foot and ankle”

72: should be “and deformities”

102: space before “years”

Results:

169: remove and replace obviously with formal language

Table 1: improve formatting for reader

189, Reference 14: this is a study on a pediatric population, however this study has a mean age of approx. 45, please find literature that represents the patients in your study.

198: change comma after “locking” to ; or –

199-203: run on sentence, please split into 2 sentences. “Additionally” should be the start of the second sentence. And please make sure that there are citations for the claims made in the first sentence.

219-220: same claim as 189. See above.

234: change talus to tali

235: be more specific than “skeletal shapes”. The shapes of which bones? What is the shape difference? What about the bone micro-architecture is different? Or remove sentence entirely.

Reviewer #2: (No Response)

7. PLOS authors have the option to publish the peer review history of their article (what does this mean?). If published, this will include your full peer review and any attached files.

Reviewer #1: No

Reviewer #2: **Yes: **Mohammad Poursalehian

---

## [Editor Report · Decision Letter 2]

11 Mar 2024

Evaluation of the horizontal approach to the medial malleolar facet in sagittal talar fractures through dorsiflexion and plantarflexion positions.

PONE-D-23-36659R2

Dear Dr. Yan,

We’re pleased to inform you that your manuscript has been judged scientifically suitable for publication and will be formally accepted for publication once it meets all outstanding technical requirements.

Kind regards,

Amir Human Hoveidaei, MD, MSc

Academic Editor

PLOS ONE

---

## [Editor Report · Acceptance letter]

30 Apr 2024

PONE-D-23-36659R2 

PLOS ONE

Dear Dr. Yan, 

I'm pleased to inform you that your manuscript has been deemed suitable for publication in PLOS ONE. Congratulations! Your manuscript is now being handed over to our production team.

Kind regards, 

on behalf of

Dr. Amir Human Hoveidaei 

Academic Editor

PLOS ONE